# Epidemiological, Clinical and Microbiological Characteristics of Patients with Bloodstream Infections Due to Carbapenem-Resistant *K.* *Pneumoniae* in Southern Italy: A Multicentre Study

**DOI:** 10.3390/antibiotics11050633

**Published:** 2022-05-08

**Authors:** Lorenzo Onorato, Bruno Sarnelli, Federica D’Agostino, Giuseppe Signoriello, Ugo Trama, Angelo D’Argenzio, Maria Vittoria Montemurro, Nicola Coppola

**Affiliations:** 1Infectious Diseases Unit, Department of Mental Health and Public Medicine, University of Campania “Luigi Vanvitelli”, 80131 Napoli, Italy; lorenzo.onorato@unicampania.it; 2Direzione Generale Tutela della Salute e Coordinamento del Sistema Sanitario Regionale della Campania, 80143 Napoli, Italy; bruno.sarnelli@regione.campania.it (B.S.); angelo.dargenzio@regione.campania.it (A.D.); 3UOSD Programmazione, Progettazione, Valutazione Strategica e Gestione della Performance-Azienda Ospedaliera “San Pio”—Benevento, 82100 Benevento, Italy; federicadagostino@live.it; 4Department of Experimental Medicine, University of Campania “Luigi Vanvitelli”, 80131 Naples, Italy; giuseppe.signoriello@unicampania.it; 5UOSD Politica del Farmaco e Dispositivi, Regione Campania, 80143 Napoli, Italy; ugo.trama@regione.campania.it; 6Medical Direction, Azienda Ospedaliera Universitaria Luigi Vanvitelli, 80138 Napoli, Italy; mariavittoria.montemurro@policliniconapoli.it

**Keywords:** Klebsiella, Carbapenem-Resistant Enterobacteriaceae, Carbapenemase-Producing Enterobacteriaceae, metallo-beta-lactamase, bloodstream infections

## Abstract

Background: In the present study, our aim was to evaluate the clinical and microbiological characteristics of a cohort of patients with bloodstream infections (BSI) due to Carbapenem-Resistant Enterobacteriaceae (CRE) and investigate the independent predictors of mortality. Methods: All episodes of carbapenem-resistant *E.* *coli* (CREc) or *K.* *pneumoniae* (CRKp) BSI that were subject to a mandatory notification from January to December 2020 in all acute care hospitals and long-term care facilities of the Campania region in southern Italy were enrolled. All carbapenem-resistant strains were assessed through molecular tests for the presence of five carbapenemase gene families, i.e., *K.* *pneumoniae* Carbapenemase (KPC), oxacillinase-48 (OXA-48), New Delhi Metallo-β-lactamase (NDM), Verona integron encoded metallo-β-lactamase (VIM) and Imipenemase (IMP). Results: During the study period, a total of 154 consecutive non-repeated CRE BSI, all due to CRKp, were reported. The most frequently identified genes were KPC in 108 cases (70.1%), followed by metallo-betalactamases (MBL) (16.2%), and OXA-48 (2.6%); in 17 isolates (11%) no carbapenemase was detected. The overall mortality at 90 days was 41.9%. Using a log-rank test, patients without risk factors for CRE infections showed a significantly lower cumulative mortality (*p* = 0.001). After multivariate logistic regression analysis, the presence of at least one risk factor was the only predictor of mortality (OR: 1.7, 95% CI 1.2–6.1, *p* = 0.015). Conclusions. The study reported a non-negligible prevalence of MBL-producing organisms among CRKp isolated from blood cultures in our region. This data highlights the importance of molecular characterization of all clinical isolates of carbapenem-resistant organisms.

## 1. Introduction

The spread of antimicrobial resistance represents a threat to global health [1]. According to the most recent data from the European Antimicrobial Resistance Surveillance Network (EARS-Net) database, a quarter of European Union (EU) countries reported carbapenem resistance rates higher than 10% in *K. pneumoniae* isolates, with peaks of 29.5% in Italy, 48.3% in Romania, and 66.3% in Greece [2]. According to the estimates based on the surveillance data, in 2015, approximately 10,000 deaths occurred in Europe due to carbapenem-resistant organisms, and about half of them were in Italy alone [3]. Not surprisingly, the world health organization (WHO) has included Carbapenem-Resistant Enterobacteriaceae (CRE) on the priority list of pathogens for which we urgently need novel treatment strategies [4].

In recent years, new beta-lactam/beta-lactamase inhibitor combinations have been developed against carbapenem-resistant Gram-negative bacteria [5]; however, their spectrum of activity is restricted to strains producing specific classes of carbapenemases. Therefore, the molecular characterization of carbapenem-resistant organisms is becoming of utmost importance in order to choose the correct treatment regimen. Unfortunately, data on the molecular epidemiology of CRE worldwide are fragmentary, and unavailable for some geographical areas [6].

In the present paper, we aimed to evaluate the clinical and microbiological characteristics of a cohort of patients with bloodstream infections (BSI) due to CRE to analyze the clinical and epidemiological factors correlated with the molecular characterization of the isolates and to investigate the independent predictors of mortality.

## 2. Methods

### 2.1. Study Design

In Italy, all bloodstream infections due to carbapenem-resistant *E. coli* (CREc) or *K. pneumoniae* (CRKp) are subject to mandatory notification to national and local authorities. In this study, we retrospectively included all episodes of CREc or CRKp BSI that were reported in the Campania Region from January to December 2020 in one of the regional acute care hospitals or long-term care facilities (LTCF). Campania has a population of almost 6 million people, with 49 acute care hospitals and 24 LTCFs. A bloodstream infection was defined as the growth from 1 or more blood cultures of a *K. pneumoniae* or *E. coli* isolate displaying resistance in vitro to 1 or more carbapenems.

For each patient enrolled, we collected the demographic data, including age and gender, and clinical data, including the site of acquisition (i.e., community-acquired or nosocomial infection), source of infection, risk factors for infection due to multidrug-resistant (MDR) Enterobacteriaceae, treatment received and 7-day and 90-day mortality. Risk factors for infection due to MDR-enterobacteriaceae included previous rectal colonization, hospital admission in the last 90 days, current admission to an intensive care unit (ICU), oncology, hematology, spinal unit or transplant surgery unit, dialysis or anti-neoplastic treatment in the last 12 months.

### 2.2. Microbiology

Bacterial identification and susceptibility testing were performed using automated commercial assays. The microbiological tests were not conducted in a centralized laboratory, so different methodologies have been used in the centers included in the study. The systems most commonly used for bacterial identification included the MALDI-TOF MS (Bruker Daltonik, Bremen, Germany) and the BD Phoenix^TM^ M50 System (Becton Dickinson). For antimicrobial susceptibility testing, most laboratories adopted the BD Phoenix^TM^ 100 system or the Vitek2 System (bioMérieux, Florence, Italy). The isolates were defined as carbapenem-resistant if they presented a non-susceptibility to at least one carbapenem among imipenem, meropenem or ertapenem according to the EUCAST clinical breakpoints, v 10.0. All carbapenem-resistant strains were assessed for the presence of five carbapenemase gene families, i.e., *K. pneumoniae* Carbapenemase (KPC), oxacillinase-48 (OXA-48), New Delhi metallo-β-lactamase (NDM), Verona integron encoded metallo-β-lactamase (VIM) and Imipenemase (IMP), through PCR tests. The kits most commonly used for the molecular assay included the Check-Direct CPE kit (Check-Points) and the Xpert Carba-R kit (Cepheid, Sunnyvale, CA, USA). If genotypic tests were not available, the production of Ambler Class A or Class B carbapenemases was assessed through the phenotypic double-disk test with boronic acid and ethylenediaminetetraacetic acid (EDTA), respectively.

### 2.3. Statistical Analysis

Continuous variables were summarized as mean and standard deviation, and categorical variables as absolute and relative frequencies. For continuous variables, the differences were evaluated by Student’s t-test; categorical variables were compared by the chi-squared test, using exact procedures if needed; 90-day survival among groups was compared using a log-rank test. To identify independent predictors of 90-day mortality, a multivariate analysis using logistic regression was performed; age, gender and variables associated with 90-day mortality using univariate analysis with a *p* value < 0.1 were included in the model. A *p* value < 0.05 was considered statistically significant. The analysis was performed using IBM SPSS v 21.0 (Armonk, NY, USA).

### 2.4. Ethics Statement

The study was conducted according to the guidelines of the Declaration of Helsinki, and approved by the Ethics Committee of the University of Campania L. Vanvitelli, Naples (N° 13255/2022)

## 3. Results

### 3.1. Demographic and Clinical Characteristics of Patients

During the study period, a total of 154 consecutive non-replicate bloodstream infections due to carbapenem resistant *K. pneumoniae* were included. No carbapenem-resistant *E. coli* BSI were reported in 2020.

The baseline characteristics of patients enrolled are summarized in Table 1. The subjects included showed a mean age of 60.8 (±18.9) and were prevalently males (63%); 136 out of 154 patients (88.3%) presented a nosocomial infection, 2 patients (1.3%) acquired the infection in an LTCF, whereas the remaining 16 patients (10.4%) showed a community-acquired infection. Almost half of the patients presented a primary or central venous catheter (CVC)-related bacteremia; other common sources included lower respiratory tract infection (18.8%), urinary tract infection (12.3%), abdominal infections (7.1%), and skin and soft tissue infections (5.2%). A total of 92 out of 154 patients (59.6%) presented a risk factor for infection due to CRE, including previous rectal carriage (11%), hospital admission during the last 90 days (14.3%), current hospitalization in a high-risk unit (31.2%), and dialysis or anti-neoplastic treatment in the last 12 months (3.2%).

The most commonly identified carbapenemase genes were KPC in 108 cases (70.1%), followed by metallo-beta-lactamases in 25 (16.2%), including 17 VIM (11%), 2 NDM (1.3%), 1 IMP (0.6%) and 5 non-specified and OXA-48 in 4 (2.6%); in 17 isolates (11%), no carbapenemase genes were detected.

### 3.2. Characteristics of Patients According to the Carbapenemase Detected

Table 2 shows the characteristics of patients according to the presence and type of carbapenemase detected. No significant differences in demographics or clinical characteristics were observed among the groups. Patients infected with MBL-producing strains were more frequently colonized at admission (15 out of 25, 60%) compared with those infected by KPC- or OXA-producing bacteria and non-Carbapenemase-producing organisms (38.4% and 23.5%, respectively).

### 3.3. Characteristics of Patients According to 7-Day Mortality

The overall mortality at 7 days from the index culture was 32.5% (50 out of 154). The characteristics of subjects stratified according to 7-day mortality are shown in Table 3.

The patients who did not survive were more frequently males (74.0% vs. 57.7%, *p* = 0.049) and presented a higher prevalence of risk factors for infections due to CRE, including previous rectal colonization and hospital admission in the last 90 days (22.0% vs. 5.8% and 20.0% vs. 11.5%, respectively; *p* = 0.003). Mean age and other clinical and microbiological factors, including type of acquisition, site of infection, carbapenemase gene detected and treatment received did not differ significantly among the groups (Table 3).

### 3.4. Characteristics of Patients According to 90-Day Mortality

Data on 90-day mortality were available for 148 (96.1%) patients; 62 (41.9%) patients died during the follow-up (Table 4).

In addition, risk factors for CRE infections were more frequently observed in the group that did not survive, such as rectal colonization (17.7% vs. 7.5%) and first admission to a high-risk unit (40.3% vs. 25.6%). Using a log-rank test, patients with risk factors showed a higher cumulative mortality rate at 90 days (*p* = 0.001, see Figure 1). Other demographic, clinical and microbiological characteristics were equally distributed among the groups.

### 3.5. Independent Predictors of 90-Day Mortality

Multivariate logistic regression analysis indicated the presence of at least one risk factor for CRE infection and was the only variable independently associated with mortality at 90 days (OR 2.71, 95% CI 1.21–6.07, *p* value 0.015, see Table 5).

## 4. Discussion

In the present study, we analyzed the demographic, clinical and microbiological characteristics of 154 consecutive patients with bloodstream infections due to carbapenem resistant K. pneumoniae. As expected, the vast majority of patients (136 out of 154, 88%) presented a nosocomial infection, and approximately 60% of subjects included showed at least one known risk factor for infection due to MDR organisms. These results are in agreement with the data available in the literature. In a recent systematic review [7], including 78 case-control and 14 cohort studies that reported data on risk factors associated with infections due to carbapenem-resistant Gram-negative pathogens among hospitalized patients, previous colonization was significantly associated with infection in 8 out of 11 studies (72.7%), ICU stays in 38 out of 59 studies (64.4%), dialysis in 11 of 18 studies (61.1%), and previous hospitalization in 15 of 41 studies (36.6%).

Regarding the microbiological characteristics, in the present study 70% of patients were infected with a KPC-producing strain. According to data from the SENTRY study [8], including 1298 CRE isolates collected in 199 hospitals from 42 countries across Europe, North and Latin America and Asia over 20 years, reported a prevalence of KPC of 54.2%, whereas MBL was isolated in 12.7% cases and OXA-48 in 12.6%, mainly in Europe. However, the molecular epidemiology of carbapenem-resistant Enterobacterales largely varies among geographical areas. The prevalence of production of carbapenemases among CRE is reported to be 35–60% in the US [9,10], with KPC being the most frequently detected gene. Class A carbapenemases are by far the most frequently isolated in several other areas, including Latin America, Greece and Israel [11], whereas OXA-48 is most commonly reported in Spain [12]. Regarding our country, a survey published in 2017, including 22 Italian sentinel hospitals, reported that among 195 carbapenem-resistant *K. pneumoniae* strains collected from November 2013 to April 2015, 95.9% of isolates tested positive for the KPC gene [13]. Similarly, in a more recent prospective study, including 691 patients with bloodstream infections due to carbapenem-resistant *K. pneumoniae* admitted to 19 hospitals in southern Italy, KPC was identified in 95.6% of the isolates, whereas VIM was isolated in 3.5% of cases [14]. Compared with these data, our study reported a higher prevalence of strains producing metallo-β-lactamases (16.2%). This finding has important implications in clinical practice, considering that the treatment options currently available against MBL-producing Gram-negative rods are limited.

When analyzing the risk factors for CRE infection, we found that patients with BSI due to MBL-producing strains presented more frequently with rectal colonization at hospital admission. Conversely, a recent prospective cohort study conducted in Singapore among 133 CRE-colonized patients reported that subjects colonized with OXA-producing strains had a higher risk of developing a subsequent infection compared with those colonized by KPC or NDM-producing Enterobacterales (OR 9.4; 95% CI 1.9–45.7; *p* = 0.02) [15]. In the same year, a cohort study enrolling 153 Chinese patients hospitalized in an ICU or hematology department with a positive rectal swab for CRE, reported that the occurrence of a subsequent infection was significantly more frequent among subjects colonized with a KPC-CRE compared with those colonized with an NDM-producing or a non-CP-CRE [16]. These mixed findings might be due to the different local epidemiology, different study design or clinical and microbiological characteristics of the patients included.

The mortality of patients with BSI by carbapenem-resistant *K. pneumoniae* included in the present study was very high: 32.5% at day 7 and 41.9% at day 90 from the index culture. This high mortality rate may be due not only to the severity of these infections but also to the fact that they more frequently involved patients at high risk of mortality (hospitalized patients with chronic underlying diseases). Regarding the variable associated with mortality, the only independent predictor found in our study was the presence of at least one risk factor for CRE infection, including previous rectal colonization, recent hospitalization, and admission to high-risk units, such as ICUs. Several studies have demonstrated that patients with bloodstream infections due to CRE presenting with sepsis or septic shock [17,18] or admitted to ICUs [19,20] showed an obviously higher mortality rate than non-critically ill patients. An additional feature frequently associated with the mortality of patients with BSI due to carbapenem resistant Enterobacteriaceae was the source of infection; urinary tract infections (UTI) have been associated to significantly reduced mortality in many studies [17,18,19]. In our cohort, the prevalence of UTI were higher among patients who survived at 90 days compared with those who did not (16.3 vs. 8.1%), but this difference was not statistically significant. However, we should consider that the number of UTI were limited in our population (19 out of 154 patients), whereas primary or CVC-related bacteremia accounted for almost half of the cases. In 2016, Gutierrez-Gutierrez [17] proposed a score to predict 14-day mortality, based on the data from a large multinational retrospective cohort including 468 patients with BSI due to Carbapenemase-Producing Enterobacteriaceae (CPE). The score was subsequently validated in several retrospective and prospective evaluations [18,21,22,23]. Interestingly, the type of carbapenemase produced was not identified as an independent predictor of mortality, which also occurred in our study.

Finally, the correlation between the molecular characterization of Carbapenem-Resistant Enterobacteriaceae and the clinical outcome of patients is still a matter of debate. In 2017, the retrospective study conducted by Tamma et al. [24] among 83 episodes of CRE bacteremia showed that patients infected with non-CPE had a significantly lower 14-day mortality compared with those infected by a carbapenemase-producing strain. Conversely, in a recently published cohort study [25], non-CP CRE bacteremia was associated with a 2.4-fold higher hazard of death at 30 days compared with CP-CRE. Similar findings were observed in a case-control study [26] enrolling 447 patients hospitalized between 2016 and 2018; the 149 subjects colonized with non-carbapenemase producing CRE presented a higher rate of mechanical ventilation and hospital mortality compared with both CPE-colonized and non-CRE colonized patients. These differences in the studies could be due to different local epidemiology and clinical characteristics of patients, as well as the availability in recent years of novel effective treatments against carbapenemase-producing microorganisms.

The strengths of our study include the large sample of bloodstream infections evaluated and the participation of several centers covering our entire geographic area, which allows us to have a clear picture of the molecular epidemiology of CRE infections in our region. However, some limitations should be highlighted. Because of the retrospective design of the study, it was not possible to retrieve much useful information, including the clinical severity of patients and the treatment received, which was only available for about 50% of cases. Moreover, some important microbiological data are lacking; firstly, the results of susceptibility testing were only available for carbapenems; no data were provided regarding the resistance rate to other antimicrobial classes, and this does not allow the evaluation of potential alternative treatments. Finally, the genome sequencing of the isolates was not performed; therefore, information about the clonal complexes of the strains was not available; similarly, we were not able to characterize the allelic variants of the carbapenemase genes, which could encode for enzymes with a different spectrum of activity.

## 5. Conclusions

The present study reports a non-negligible prevalence of metallo-beta-lactamases producing strains among 154 consecutive carbapenem-resistant *K. pneumoniae* isolates causing bloodstream infections in Campania in 2020. These data highlight the importance of molecular characterization of all carbapenem-resistant pathogens in order to select the optimal treatment regimen and improve patient outcomes.

## Figures and Tables

**Figure 1 antibiotics-11-00633-f001:**
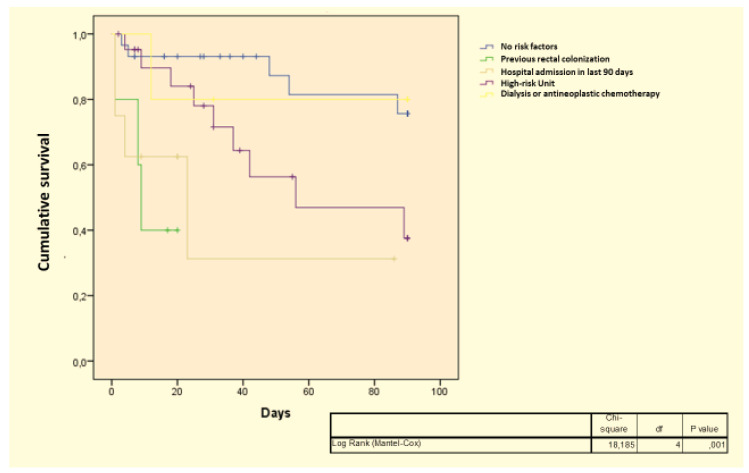
Kaplan–Meier curve on the cumulative mortality rate at 90 days according to risk factors.

**Table 1 antibiotics-11-00633-t001:** Demographic, clinical and microbiological characteristics of the patients enrolled.

N° of patients	154
Mean age (SD), years	60.8 (18.9)
Males, n° (%)	97 (63.0)
Acquisition, n° (%)	
- Community - Nosocomial - ICU - Medical ward - Surgical ward - Unknown - Long-term care facility	16 (10.4) 136 (88.3) 78 (50.6) 28 (18.2) 18 (11.7) 12 (7.8) 2 (1.3)
Carbapenemase detection, n° (%)	
- KPC/OXA-48 - Metallo-beta-lactamases - None	112 (72.7) 25 (16.2) 17 (11.0)
Site of infection	
- Primary/CVC-related BSI - HAP/VAP - UTI - Intra-abdominal infection - Skin and soft tissue - Multiple sites - Others	73 (47.4) 29 (18.8) 19 (12.3) 11 (7.1) 8 (5.2) 4 (2.6) 10 (6.5)
Risk factors	
- Previous rectal colonization - Hospital admission in the last 90 days - First admission to high-risk unit * - Dialysis or anti-neoplastic chemotherapy in the last 12 months - None	17 (11.0) 22 (14.3) 48 (31.2) 5 (3.2) 62 (40.3)
Antibiotic treatment	
- CAZ/AVI monotherapy - CAZ/AVI combination therapy - Colistin-based regimens - Others - Unknown	39 (25.3) 12 (7.8) 8 (5.2) 15 (9.7) 80 (51.9)
7-day mortality	50 (32.5)
90-day mortality **	62 (41.9)

Footnotes: ICU: intensive careu, KPC: Klebsiella pneumoniae carbapenemase; OXA-48: Oxacillinase-48; CVC: central venous catheter, BSI: bloodstream infection, HAP: hospital-acquired pneumonia, VAP: ventilator-associated pneumonia; UTI: urinary tract infection, CAZ/AVI: Ceftazidime/avibactam; * high-risk units include: ICU, oncology, hematology, spinal units, transplant surgery; ** data available for 148 patients.

**Table 2 antibiotics-11-00633-t002:** Demographic, clinical and microbiological characteristics of the patients according to the carbapenemase detected.

	MBL	KPC/OXA-48	No Carbapenemase Detected	*p* Value
N° of patients	25	112	17	
Mean age (SD), years	49.1 (29.5)	63.0 (15.3)	63.0 (15.9)	0.24
Males, n° (%)	13 (52.0)	74 (66.1)	10 (58.8)	0.39
Acquisition, n° (%)				
- Community - Nosocomial - ICU - Medical ward - Surgical ward - Unknown - Long-term care facility	0 (0.0) 24 (96.0) 18 (72.0) 5 (20.0) 0 (0.0) 1 (4.0) 1 (4.0)	15 (13.4) 96 (85.7) 49 (43.7) 23 (20.5) 16 (14.3) 8 (7.1) 1 (0.9)	1 (5.9) 16 (94.1) 12 (70.6) 0 (0.0) 3 (17.6) 1 (5.9) 0 (0.0)	0.20
Site of infection				
- Primary/CVC-related BSI - HAP/VAP - UTI - Intra-abdominal infection - Skin and soft tissue - Multiple sites - Others	11 (44.0) 4 (16.0) 5 (20.0) 0 (0.0) 2 (8.0) 1 (4.0) 2 (8.0)	57 (50.9) 16 (14.3) 12 (10.7) 11 (9.8) 6 (5.4) 3 (2.7) 7 (6.2)	5 (29.4) 9 (52.9) 2 (11.8) 0 (5.9) 0 (0.0) 0 (0.0) 1 (5.9)	0.052
Risk factors				
- Previous rectal colonization - Hospital admission in the last 90 days - First admission to high-risk unit * - Dialysis or anti-neoplastic chemotherapy in the last 12 months - None	15 (60.0) 0 (0.0) 2 (8.0) 8 (32.0) 0 (0.0)	43 (38.4) 17 (15.2) 17 (15.2) 30 (26.8) 5 (4.5)	4 (23.5) 0 (0.0) 3 (17.6) 10 (58.8) 0 (0.0)	**0.024**
Antibiotic treatment				
- CAZ/AVI monotherapy - CAZ/AVI combination therapy - Colistin combination - Others - Unknown	0 (0.0) 0 (0.0) 2 (8.0) 6 (24.0) 17 (68.0)	38 (33.9) 9 (8.0) 1 (0.9) 6 (5.3) 58 (51.8)	1 (5.9) 3 (17.6) 5 (29.4) 3 (17.6) 5 (29.4)	**<0.001**
7-day mortality	4 (16.0)	41 (36.6)	5 (29.4)	0.13
90-day mortality **	6 (24.0)	47 (43.9)	9 (56.2)	0.09

Footnotes: MBL: mtallo-beta-lactamase; KPC: Klebsiella pneumoniae carbapenemase; OXA-48: Oxacillinase-48; ICU: intensive care unit; CVC: central venous catheter, BSI: bloodstream infection, HAP: hospital-acquired pneumonia, VAP: ventilator-associated pneumonia; UTI: urinary tract infection, CAZ/AVI: ceftazidime/avibactam; * high-risk units include: ICU, oncology, hematology, spinal units, transplant surgery; ** data available for 132 patients; *p* values < 0.05 are displayed in bold.

**Table 3 antibiotics-11-00633-t003:** Demographic, clinical and microbiological characteristics of patients according to 7-day mortality.

	Survivors	Dead	*p* Value
N° of patients	104	50	
Mean age (SD), years	59.6 (20.0)	63.1 (16.6)	0.26
Males, n° (%)	60 (57.7)	37 (74.0)	**0.049**
Acquisition, n° (%)			
- Community - Nosocomial - ICU - Medical ward - Surgical ward - Unknown - Long-term care facility	10 (9.6) 93 (89.4) 48 (46.1) 23 (22.1) 13 (12.5) 9 (5.8) 1 (0.9)	6 (12.0) 43 (86.0) 30 (60.0) 5 (10.0) 5 (10.0) 3 (6.0) 1 (2.0)	0.77
Carbapenemase detection, n° (%)			
- KPC/OXA-48 - Metallo-beta-lactamases - None	71 (68.3) 21 (20.2) 12 (11.5)	41 (82.0) 4 (8.0) 5 (10.0)	0.13
Site of infection			
- Primary/CVC-related BSI - HAP/VAP - UTI - Intra-abdominal infection - Skin and soft tissue - Multiple sites - Others	46 (44.2) 21 (20.2) 15 (14.4) 6 (5.8) 5 (4.8) 2 (1.9) 9 (8.6)	27 (54) 8 (16) 4 (8) 5 (10) 3 (6) 2 (4) 1 (2)	0.57
Risk factors			
- Previous rectal colonization - Hospital admission in the last 90 days - First admission to high-risk unit * - Dialysis or anti-neoplastic chemotherapy in the last 12 months - None	6 (5.8) 12 (11.5) 31 (29.8) 4 (3.8) 51 (49.0)	11 (22) 10 (20) 17 (34) 1 (2) 11 (22)	**0.003**
Antibiotic treatment			
- CAZ/AVI monotherapy - CAZ/AVI combination therapy - Colistin-based regimens - Others - Unknown	31 (29.8) 7 (6.7) 6 (5.8) 11 (10.6) 49 (47.1)	8 (16.0) 5 (10.0) 2 (4.0) 4 (8.0) 31 (62.0)	0.29

Footnotes: ICU: intensive care unit, KPC: Klebsiella pneumoniae carbapenemase; OXA-48: Oxacillinase-48; CVC: central venous catheter, BSI: bloodstream infection, HAP: hospital-acquired pneumonia, VAP: ventilator-associated pneumonia; UTI: urinary tract infection, CAZ/AVI: ceftazidime/avibactam; * high-risk units include: ICU, oncology, hematology, spinal units, and transplant surgery; p values < 0.05 are displayed in bold.

**Table 4 antibiotics-11-00633-t004:** Demographic, clinical and microbiological characteristics of patients according to 90-day mortality.

	Survivors	Dead	*p* Value
N° of patients	86	62	
Mean age (SD), years	58.2 (20.9)	62.7 (15.7)	0.14
Males, n° (%)	50 (58.1)	44 (71.0)	0.11
Acquisition, n° (%)			
- Community - Nosocomial - ICU - Medical ward - Surgical ward - Unknown - Long-term care facility	9 (10.5) 76 (88.4) 38 (44.2) 20 (23.2) 11 (12.8) 7 (8.1) 1 (1.2)	6 (9.7) 55 (88.7) 39 (62.9) 7 (11.3) 5 (8.1) 4 (6.4) 1 (1.6)	0.96
Carbapenemase detection, n° (%)			
- KPC/OXA-48 - MBL - None	60 (69.8) 19 (22.1) 7 (8.1)	47 (75.8) 6 (9.7) 9 (14.5)	0.09
Site of infection			
- Primary/CVC-related BSI - HAP/VAP - UTI - Intra-abdominal infection - Skin and soft tissue - Multiple sites - Others	38 (44.2) 13 (15.1) 14 (16.3) 6 (7.0) 4 (4.7) 2 (2.3) 9 (10.5)	32 (51.6) 14 (22.6) 5 (8.1) 5 (8.1) 3 (4.8) 2 (3.2) 1 (1.6)	0.25
Risk factors			
- Previous rectal colonization - Hospital admission in the last 90 days - First admission to high-risk unit * - Dialysis or anti-neoplastic chemotherapy in the last 12 months - None	3 (3.5) 10 (11.6) 22 (25.6) 4 (4.7) 47 (54.7)	11 (17.7) 10 (16.1) 25 (40.3) 1 (1.6) 15 (24.2)	**0.001**
Antibiotic treatment			
- CAZ/AVI monotherapy - CAZ/AVI combination therapy - Colistin based combination - Others - Unknown	23 (26.7) 5 (5.8) 5 (5.8) 11 (12.8) 42 (48.8)	13 (21.0) 7 (11.3) 3 (4.8) 4 (6.5) 35 (56.5)	0.44

Footnotes: ICU: intensive care unit, KPC: Klebsiella pneumoniae carbapenemase; OXA-48: Oxacillinase-48; CVC: central venous catheter, BSI: bloodstream infection, HAP: hospital-acquired pneumonia, VAP: ventilator-associated pneumonia; UTI: urinary tract infection, CAZ/AVI: ceftazidime/avibactam; * high-risk units include: ICU, oncology, hematology, spinal units, and transplant surgery. *p* values < 0.05 are displayed in bold.

**Table 5 antibiotics-11-00633-t005:** Independent predictors of 90-day mortality at multivariate logistic regression analysis.

	OR	95% Confidence Interval		*p* Value
		Lower	Upper	
Age	1.014	0.992	1.036	0.214
Sex (M vs. F)	1.757	0.766	4.033	0.184
No carbapenemase detected (ref.)				
- KPC or OXA-48 - MBL	0.53 0.369	0.163 0.081	1.717 1.68	0.289 0.197
Presence of risk factors	2.715	1.215	6.068	**0.015**

Footnotes: KPC: Klebsiella pneumoniae carbapenemase; OXA-48: Oxacillinase-48; MBL: metallo-beta-lactamase, *p* values < 0.05 are displayed in bold.

## Data Availability

The data presented in this study are available on request from the corresponding author.

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
