# Peer review of "Epidemiological, Clinical and Microbiological Characteristics of Patients with Bloodstream Infections Due to Carbapenem-Resistant K. Pneumoniae in Southern Italy: A Multicentre Study"

_antibiotics, 2022, doi:10.3390/antibiotics11050633_

Round 1

Reviewer 1 Report

This article presents a very serious worldwide problem concerning CRE. More spesific the authors. More specific the authors investigate the e clinical and
microbiological characteristics of a cohort of patients with bloodstream infections (BSI) due to carbapenem-resistant Enterobacteriaceae (CRE) from a  multicenter Italian study.The methods are not so sufficient explainds I believe they can be imporved like what kind of laboratory assays have been used. The results are clear and well documented.The arrangement of the figure and tabels is clear and presents the obtained results very well.  The publications cited by the authors of the article are well selected. 

I believe it is an interesting article and I recommend that it could be published with minor revision concerning the Methods.

Author Response

Dear Editor,

We re-submit our paper “Epidemiological, clinical and microbiological characteristics of patients with bloodstream infections due to carbapenem-resistant K. pneumoniae in Southern Italy: a multicentre study (tracking number: antibiotics-1687507), modified according to the suggestions of the reviewers.

POINT-BY-POINT ANSWER TO THE COMMENTS OF THE REVIEWER 1

Point 1: The methods are not so sufficient explained I believe they can be improved like what kind of laboratory assays have been used.

Answer: We added in the Methods section further data on the microbiological methodology used in the study (lines 91-95 and 100-102).

We thank the Reviewers and the Editor for helping us to improve our paper.

We hope that the paper is now worthy of publication in Antibiotics

Best regards,

Prof Nicola Coppola

Reviewer 2 Report

This is a well structured and relevant piece of work. The only comment I would like to make is around Table 3 and Table 4, I believe that 7-day and 90-day mortality will be better explored by using a univariate and multivariate cox-regression analysis for each variable. 

Author Response

Dear Editor,

We re-submit our paper “Epidemiological, clinical and microbiological characteristics of patients with bloodstream infections due to carbapenem-resistant K. pneumoniae in Southern Italy: a multicentre study (tracking number: antibiotics-1687507), modified according to the suggestions of the reviewers.

POINT-BY-POINT ANSWER TO THE COMMENTS OF THE REVIEWER 2

Point 1 The only comment I would like to make is around Table 3 and Table 4, I believe that 7-day and 90-day mortality will be better explored by using a univariate and multivariate cox-regression analysis for each variable

Answer: The impact of the different variables on the mortality outcome has already been explored through univariate and multivariate analysis using a logistic regression model (see Results, lines 160-162), and independent predictors of mortality have been identified. Thus, we believe that performing an additional multivariate analysis with a different statistical method could be redundant and potentially misleading. However, if the reviewer retains that this method will provide additional information, we will rerun the analysis using a Cox regression model.

We thank the Reviewers and the Editor for helping us to improve our paper.

We hope that the paper is now worthy of publication in Antibiotics

Best regards,

Prof Nicola Coppola

Reviewer 3 Report

The study is clearly presented and the manuscript reads well. I will suggest that the authors correct K. pneumoniae in their title (ln 4), as well as to put all the bacterial names mentioned in the main body of the text in italics. 

Author Response

Dear Editor,

We re-submit our paper “Epidemiological, clinical and microbiological characteristics of patients with bloodstream infections due to carbapenem-resistant K. pneumoniae in Southern Italy: a multicentre study (tracking number: antibiotics-1687507), modified according to the suggestions of the reviewers.

POINT-POINT ANSWER TO THE COMMENTS OF THE REVIEWER 3

Point 1 I will suggest that the authors correct K. pneumoniae in their title (ln 4), as well as to put all the bacterial names mentioned in the main body of the text in italics.

Answer: As suggested by the reviewer, we wrote all the bacterial names in italics

We thank the Reviewers and the Editor for helping us to improve our paper.

We hope that the paper is now worthy of publication in Antibiotics

Best regards,

Prof Nicola Coppola

Reviewer 4 Report

The Authors described the epidemiological, clinical and microbiological characteristics of patients with bloodstrem infections due to carbapenem-resistant Klebsiella pneumoniae in Southern-Italy. The manuscript prensents interesting data updating the local epidemiology. However, several suggestions need to be addressed in order to improve its quality:

-the microbiological section has been poorly described: the methods for identification, phenotypic testing and molecular detection of carbapenemases is not well detailed and should be better described.

-there are no results of antimicrobial susceptibility testing and this negatively impacts clinical data interpretation.

-the strains lack clonality characterization and carbapenemase allelic variant determination. These should be included among the study limitations

- the references should be expanded to include additional epidemiological data

Author Response

Dear Editor,

We re-submit our paper “Epidemiological, clinical and microbiological characteristics of patients with bloodstream infections due to carbapenem-resistant K. pneumoniae in Southern Italy: a multicentre study (tracking number: antibiotics-1687507), modified according to the suggestions of the reviewers.

POINT-POINT ANSWER TO THE COMMENTS OF THE REVIEWER 4

Point 1 The microbiological section has been poorly described: the methods for identification, phenotypic testing and molecular detection of carbapenemases is not well detailed and should be better described.

Answer: As suggested by the reviewer, we reported in the text additional information on the microbiological methods used for the identification and molecular characterization of the strains included in the analysis (lines 91-95 and 100-102).

Point 2 There are no results of antimicrobial susceptibility testing and this negatively impacts clinical data interpretation.

Answer: Thanks for the comment. Unfortunately, because of the retrospective design of the study some important clinical and microbiological information, such as the clinical severity of infections as well as the pattern of susceptibility of isolates were not available and could not be retrieved. We added these limitations in the discussion (lines 235-243).

Point 3 The strains lack clonality characterization and carbapenemase allelic variant determination. These should be included among the study limitations.

Answer: As reported in the previous point, we added also these limitation in the discussion (lines 243-246)

Point 4 The references should be expanded to include additional epidemiological data

Answer: As requested, additional references on the molecular epidemiology of CRE have been added in the manuscript. (See lines 176-184 and References 8-12)

We thank the Reviewers and the Editor for helping us to improve our paper.

We hope that the paper is now worthy of publication in Antibiotics

Best regards,

Prof Nicola Coppola

Round 2

Reviewer 4 Report

The Authors succesfully addressed all concerns addressed by the Reviewer, even if some limitations could not be implemented in the manuscript due to the lack of data.